

# Speleothem stable isotope reference records for East-Central
# Europe - Resampling sedimentary proxy records to get evenly
# spaced time-series with spectral control
István Gábor Hatvani[1*], Zoltán Kern[1], Szabolcs Leél-Őssy[2], Attila Demény[1]
[1]Institute for Geological and Geochemical Research, Research Centre for Astronomy and Earth Sciences, Hungarian
Academy of Sciences, Budapest, Budaörsi út 45, H-1112, Hungary
[2]Eötvös Loránd University, Department of Physical and Applied Geology, Budapest, Pázmány Péter stny. 1/C, H-1117,
Hungary
*Correspondence to: István Gábor Hatvani (hatvaniig@gmail.com)
**Abstract.** Uneven spacing is a common feature of sedimentary paleoclimate records, in many cases causing difficulties
in the application of classical statistical and time series methods. Although special statistical tools do exist to assess
unevenly spaced data directly, the transformation of such data to a temporarily equidistant time series applicable to
commonly used statistical tools remains, however, an unachieved goal. The present paper, therefore, introduces an
approach to obtain evenly spaced time series (using cubic spline fitting) from unevenly spaced speleothem records with
the application of a spectral control to avoid spectral bias caused by interpolation and retain the original spectral
characteristics of the data. The methodology was applied to stable carbon and oxygen isotope records derived from two
stalagmites of the Baradla Cave (NE Hungary) dating back to the late 18th century; it was also applied to additional well-
dated and high-resolution stable isotope records from the Han-sur-Lesse Cave (Belgium). To show the benefit of these
equally spaced records to climate studies, their coherence with primary and complex climate indices is explored using
wavelet transform coherence and discussed. The results shed light on clear relationships with climate and NAO indices,
lending support to the approach utilized in this study. Moreover, these suggest that the Baradla composite stable isotope
data can serve as regional reference records for the past ~200 years. The equally spaced time series obtained, are available
at doi: 10.1594/PANGAEA.875917.

1. **Introduction**
With more than a hundred speleothem studies published every year, it is trivial to state that speleothems are one of the
most important objects of paleoclimate research. Compared to other continental carbonate deposits, they are especially
valuable since (i) they are formed in relatively protected cave environments that render late-stage alteration less frequent,
(ii) they can be dated using numerical dating methods (e.g. U-Th series and radiocarbon dating), (iii) they can be sampled
at high spatial and temporal resolution, and (iv) they provide a number of data that serve as proxies for climate conditions
(e.g. textural characteristics, trace element and stable isotope compositions, for further details see the comprehensive
review by (Fairchild and Baker, 2012)).



The main difficulty in the application of speleothem compositions to paleoclimate research is the complex interplay of a
number of factors that may govern the proxy data and the backgrounds and roles, about which there is a lack of clarity
(see e.g. Govin et al., 2015). Global, regional and local processes collectively determine the final geochemical signature
that can be used in paleoclimate evaluation assuming that a transfer function can be established. To assist in the
interpretation of speleothem compositions, the most frequently used and widely accepted method is cave monitoring,
where the physical- and chemical parameters of the cave environment and carbonate precipitation are determined in a
multi-year study (e.g. Breitenbach et al., 2015; Mattey et al., 2008; Mattey et al., 2016; Riechelmann et al., 2011). The
advantage of this approach is the possibility of gaining direct information on cave behavior and speleothem formation
processes in the course of environmental changes; the drawback is the usually short time scale, which rarely extends as
far as a decade (Fairchild and Baker, 2012; Mattey et al., 2016). The appearance or absence of seasonality changes in the
studied cave can be determined, but the effects of stronger environmental and climate changes may not be observed due
to the short periods covered.
Another approach is the comparison of meteorological parameters and speleothem data that cover a sufficient calibration
period (e.g. Demény et al., 2017; Jex et al., 2010). If correlated with climate parameters, such data can serve as benchmark-
records. Thus, the comparison of these records may elucidate regional proxy correlations, or can be used to test complex
climate models (Wackerbarth et al., 2012). The main criteria for the selection of speleothem reference records are good
dating precision and high-resolution (close to annual). Once such a record is selected, it should be compared with
meteorological data. The comparison can be made visually, but detailed statistical data processing (e.g. Baker et al., 2015;
von Gunten et al., 2012; Wassenburg et al., 2016) can provide much more objective results.
It is a frequently-seen that data is obtained from spatially evenly sampled sedimentary sequence are smoothed with fix-
point moving average. However, if the accumulation rate was not constant over time, the smoothed curve will not provide
uniform resolution over time and cannot achieve equal spacing in time.
Unfortunately, despite all efforts, uneven spacing is a common feature of sedimentary paleoclimate records. This
characteristic usually prohibits the application of classical statistical and time series methods in many cases (Hammer,
2017; Mudelsee, 2010). There are excellent toolsets that can solve certain problems directly on unevenly spaced data, e.g.
determining whether it has a first-order autoregressive characteristic (Schulz and Mudelsee, 2002), estimating correlation
with uncertainty limits between unevenly spaced autocorrelated series (Roberts et al., 2017), or conducting a variety of
spectral analyses (Schulz et al., 1999; Schulz and Stattegger, 1997). However, the problem remains: the transformation
of unevenly spaced data to a temporally equidistant time series in order for it to be suitable for the application of
commonly used statistical tools. The required preprocessing step, in the case of sedimentary records, is most commonly
performed using linear- (e.g. Holmgren et al., 2003; Lachniet et al., 2004) or non-linear interpolation (e.g. Ersek et al.,
2012), rescaling of the data (e.g. Deininger et al., 2016) or by insufficiently documented methods (e.g. McCabe-Glynn et
al., 2013; Duan et al., 2014). However, any transformation which adds data to or removes it from the original record
unavoidably changes its spectral characteristics.
Thus, the present paper aims to introduce an approach on selected stable carbon and oxygen isotope records from the
Baradla Cave (Demény et al., 2017) to obtain evenly spaced time series from unevenly spaced speleothem records with
the application of a spectral control to avoid spectral bias and retain the genuine spectral characteristics. Section 2
describes the data sources and the proposed methodology for interpolation and resampling, combining the abilities of two
available software (Björg Ólafsdóttir et al., 2016; Schulz and Mudelsee, 2002) and the development of the spectral control.
Sect. 3 presents the evenly spaced data generated by the application of the methodology, while Sect. 4 presents an





additional application to a recently growing stalagmite from the Han-sur-Lesse Cave (Belgium; Van Rampelbergh et al.,
2015). Thus, a generally applicable statistical processing methodology to obtain spectral bias-free time series with equal
time steps and present the processed regional reference records is presented.

**2.   Materials and Methods**
**2.1.   Data sources**
The stable carbon and oxygen isotope compositions presented in this paper derive from two stalagmites (VK1 and NU2)
of the Baradla Cave (NE Hungary, N48°28′ E20°30′). A detailed description of the stalagmites and the analytical methods
can be found in (Demény et al., 2017). Conventionally, the stable C and O isotope compositions of speleothem carbonate
are expressed as $\delta^{13}C$ and $\delta^{18}O$ values in ‰. $\delta^{13}C$ or $\delta^{18}O = (R_{sa}/R_{st} - 1) \cdot 1000$, where $R_{sa}$ and $R_{st}$ are $^{13}C/^{12}C$ and $^{18}O/^{16}O$
ratios in the sample and standard (VPDB), respectively. To construct composite isotope records from the two stalagmites'
$\delta^{13}C$ and $\delta^{18}O$ records, the original raw data was normalized using mean and standard deviation for the 1950-2000 period
(an interval of simultaneous growing for the VK1 and NU2 stalagmites) and merged to a common timescale to provide a
regional reference. The composite regional reference records will be mentioned as $\Delta^{13}C_{Baradla}$ and $\Delta^{18}O_{Baradla}$ hereinafter.
To verify the applicability of the processed data the relationship of the processed -and thus evenly spaced (for details see
Section 2.1)- $\Delta^{13}C_{Baradla}$ and $\Delta^{18}O_{Baradla}$ data with climate was assessed. Thus, monthly averages of temperature and
monthly precipitation totals were retrieved, corresponding to the cave location from a global gridded climate dataset
(CRU TS3.23; Harris et al., 2014) with a time span covering 1901-2014 and a grid space of 0.5°.
Besides temperature and precipitation, indices describing the leading mode of atmospheric variability of the Atlantic
sector called North Atlantic Oscillation (NAO; Hurrell, 1995) were considered. The NAO index quantifies the pressure
difference between the Azores High and the Iceland Low regions. In a positive mode (with positive NAO indices) the
pressure difference is large and moisture and heat are transported to the northwestern part of Europe. In a negative NAO
mode (with negative NAO indices) the North Atlantic jet is shifted to the south, bringing moisture and heat to the
Mediterranean region. Since there is no unique way to define the spatial structure of the NAO, there is no universally
accepted index to describe the temporal evolution of the phenomenon (Hurrell and Deser, 2009). Most modern NAO
indices are derived either from the simple difference in surface pressure anomalies between various northern and southern
benchmark meteorological stations, or from the principal component (PC)-based time series of the leading empirical
orthogonal function of the sea level pressure (SLP) field of a selected domain.
Although PC-based indices are presumed to be more optimal representations of the full spatial patterns of the NAO and
are also presumed to be less noisy than station-based indices, they are not available as far back in time as the station-
based indices. Therefore, two datasets of monthly North Atlantic Oscillation indices ($NAO_i$) were used as a reference in
this study: (i) PC-based NAO indices ($NAO_{PC}$) calculated over the Atlantic sector (20°-80°N, 90°W-40°E) (NCARS,
2016) available from 1899 AD and (ii) station-based ones ($NAO_{st}$) obtained as monthly averages, calculated from a
quality-checked daily dataset extending back to 1850 AD constructed using mostly observational daily SLP data from
SW Iceland and the Azores; missing daily data was filled in with reanalysis products (Cropper et al., 2015).
Besides the Baradla records , stable isotope compositions published for Belgium, close to the Atlantic Ocean (Van
Rampelbergh et al., 2015), were selected and processed using the same approach (Section 2.1).

**2.2.   Methodology**

116          **2.2.1. Interpolation and resampling with spectral control**



To achieve a regular time axis, the gaps in the $\Delta^{13}C_{Baradla}$ and $\Delta^{18}O_{Baradla}$ time series were filled by cubic-spline fitting, and
resampled to an annual resolution by averaging (Fig. 1) (De Boor, 1978) in an R statistical environment (R Core Team,
2016) using the `stringr` package (Wickham, 2015) and the spline function of the `stats` package (R Core Team,
2016). The advantage of cubic spline fitting is that it is considered to be highly effective in preserving the spectral
characteristics of the original record (Horowitz, 1974), and it outperforms linear interpolation -especially in the higher
frequency domain (Schulz and Stattegger, 1997)-, and has a smaller chance of overfitting as a higher order spline. In the
case of interpolation, regardless of the applied method, the high frequency components in a spectrum will be
underestimated, thus the interpolated time series will become "reddened" (Schulz and Stattegger, 1997).

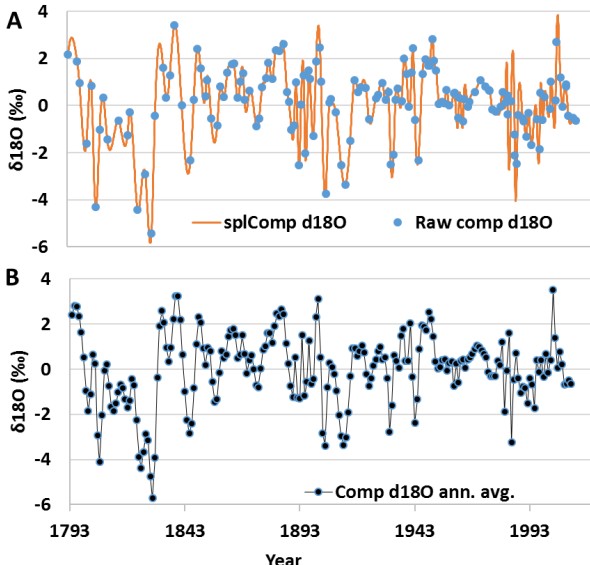


**Figure 1: Cubic spline fitted to the (a) original unevenly spaced $\Delta^{18}O_{Baradla}$ time series and the (b) annual**
**averages derived from the interpolated time series.**

Thus, the spectral bias caused by interpolation has to be objectively quantified and controlled. The spectral characteristics
of the original and the interpolated data have to be compared to detect potential spectral artifacts. Thus, the threshold
frequency was quantified beyond which (i) the interpolation (cubic spline in the present case) left the original spectrum
mainly unchanged, and (ii) the significant powers of the original time series were in coherence with the interpolated
spectrum.
To determine this threshold frequency objectively, first the spectral characteristics of the original-, and the interpolated
time series were explored to see if the powers of the time series' Lomb-Scargle Fourier Transform periodograms (LSP)
(Lomb, 1976; Scargle, 1982) are significant against red-noise background from a first-order autoregressive (AR(1))
process using REDFIT (Schulz and Mudelsee, 2002). In the calculations, a Welch I window with two overlapping (50%)
segments was used, the oversampling parameter was set as `ofac=4` according to (Hocke and Kämpfer, 2009; Press et
al., 1996), and `hifac=1`, the number of Monte-Carlo simulations to obtain the significance levels was `nsim=1000` in
line with studies dealing with speleothem time series (Holzkämper et al., 2004; Neff et al., 2001), and the red-noise





boundary was estimated as the bias-corrected 80% chi-squared limit of a fitted AR(1) process. For the computations the
`redfit` function of the `dplR` package was used (Bunn, 2008). The obtained periodograms will be referred to hereinafter
as redfit Lomb-Scargle Fourier Transform periodograms (rLSP). The obtained rLSPs of the original and the interpolated
time series were than visually compared.
After the visual comparison of the rLSPs, the coherence of the unevenly spaced original and the interpolated time series
was also computed, using REDFIT-X (Björg Ólafsdóttir et al., 2016) developed for cross-spectral analysis of unevenly
spaced paleoclimate time series. The run parameters were the same as indicated above.
Combining the visual comparison of the original and interpolated time series' rLSPs with their quantified coherence
spectrum, the smallest significant period of the original data was determined, which was also present, and in coherence
with the spline interpolated one. This could be set as the threshold frequency above which the original spectra could be
taken as unbiased. Finally, the variance below this threshold frequency was removed/filtered from the spline interpolated
time series using the bandpass function of the astrochron package (Meyers, 2014).

**2.2.2. Wavelet transform coherence (WTC)**
The data preprocessing to ensure a time series is evenly spaced in time (Section 2.2.1) was necessary to find the areas
with common powers between the speleothem stable isotope time series and the climatic data in the time-frequency space.
Wavelet transform coherence (Torrence and Webster, 1999) was used to assess and visualize the coherence of the time
series on so-called power spectrum density (PSD) graphs (e.g. Fig. 3). This method may be considered similar to a
correlation coefficient, but with the difference that here we are dealing with a localized time-frequency space (Grinsted
et al., 2004). It is based on wavelet spectrum analysis, a function localized in both frequency and time, with a mean of
zero (Grinsted et al., 2004), and may be taken as the convolution of the data and the wavelet function (Kovács et al., 2010)
for a time series ($X_n$, $n=1,\dots,N$) with a '$\Delta t$' degree of uniform resolution Eq. (1):
$$W_n^X(s) = \sqrt{\frac{\Delta t}{s}} \sum_{n'=1}^{N} X_{n'} \Psi^* \left[ (n'-n) \frac{\Delta t}{s} \right] \tag{1}$$

Whereby $N$ is the length of the time series, '$\psi^*$' is the wavelet function and '$s$' is the scale. In the study the Morlet mother
wavelet (Morlet et al., 1982) was used to generate daughter wavelets, because it establishes a clear distinction between
random fluctuations and periodic regions (Andreo et al., 2006) and has also been used in other speleothem studies (e.g.
Ersek et al., 2012; Holmgren et al., 2003).
If two time series correlate, it does not mean that their WTC will indicate a strong common periodic behavior, because
the periodic component has to be present in order to find a meaningful WTC. In the course of the evaluation only those
positive signals were considered which were significant ($\alpha=0.1$) against an AR(1) process, for details see (Roesch and
Schmidbauer, 2014). Since, the wavelet functions are normalized to have unit energy, the obtained wavelet transforms
may even be compared with other time series (Torrence and Compo, 1998).
From a practical point of view, the PSD graphs of the WTC analysis between the stable isotope time series and the
monthly climate data were calculated to find the months with the highest response. These chosen months were averaged
and their WTC with the proxy time series was calculated again. Thus, consecutive multi-monthly averages (seasons) were
obtained, indicating the maximum response forming the final output of the analysis. The WTC PSD graphs were generated
with the analyze.coherency function of the Wavelet-comp package (Roesch and Schmidbauer, 2014) in R.





## 3. Results and Discussions

### 3.1. Data preprocessing before WTC analysis

After obtaining the rLSPs and the coherency spectra of the original- and spline interpolated $\Delta^{13}C_{Baradla}$ and $\Delta^{18}O_{Baradla}$ time series, these were compared as discussed in **Section 2.2.1.**

As low as the ~5yr period, the significant powers (α=0.8) of the original $\Delta^{18}O_{Baradla}$ record were all mirrored in the spline interpolated one with high coherency and the original- and the interpolated record indicated a similar pattern on their rLSPs (Fig. 2a). However, below the ~5yr period, the significant powers of the original $\Delta^{18}O_{Baradla}$ record were no longer reflected any more in the spline interpolated one, and the two time series' coherency became generally low, as well. Therefore, to be consistent and conservative, the period domain below 4.5 yrs was omitted with a lowpass filter for the $\Delta^{18}O_{Baradla}$ time series.

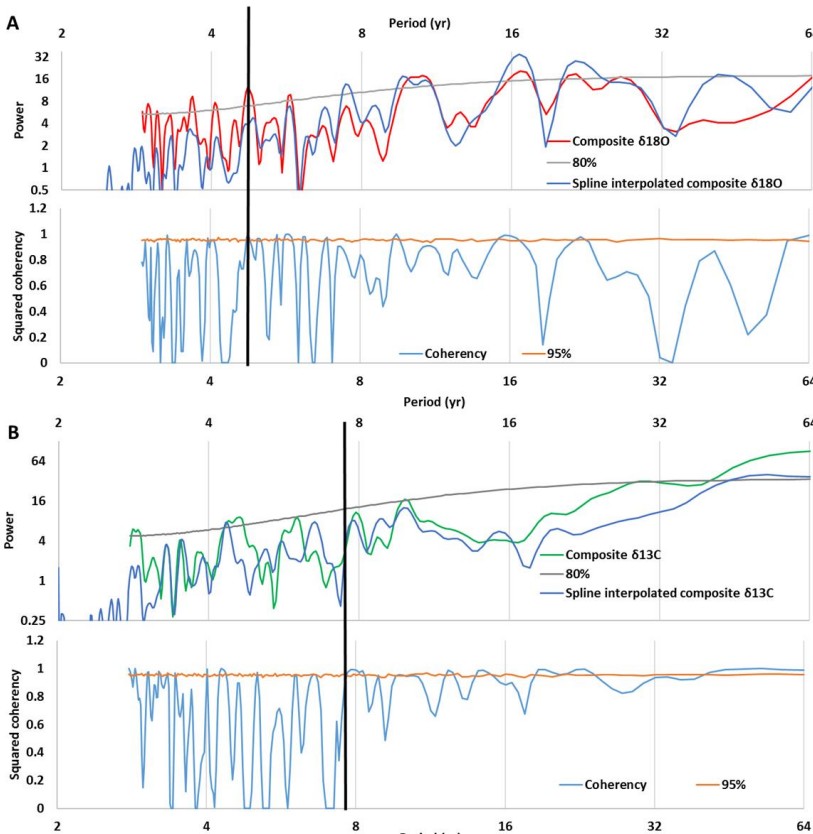

**Figure 2: rLSPs (upper panels) and coherency spectra (lower panels) of the original, and the spline interpolated (a) $\Delta^{18}O_{Baradla}$ and (b) $\Delta^{13}C_{Baradla}$ records of the Baradla speleothem. The bias-corrected 80% chi-squared limit of a fitted AR(1) process for the rLSPs is shown in grey. The coherency spectra were produced with a 95% Monte Carlo false-alarm levels (lower panels). The vertical black line indicates the determined cut-off period.**



The same steps were then performed for the $\mathit{\Delta}^{13}C_{Baradla}$ record, as well, thus the domain below 7.5yrs was cut off from the
$\mathit{\Delta}^{13}C_{Baradla}$ spectrum to avoid the spectral bias caused by spline interpolation (Fig. 2b).

**3.2.  Climate-composition relationship for the Baradla speleothems**
To present the applicability of the now equally spaced paleoproxy data in regional climate models, the $\mathit{\Delta}^{13}C_{Baradla}$ and
$\mathit{\Delta}^{18}O_{Baradla}$ records were compared to local (precipitation and temperature) and regional (NAO indices) climate variables.
The first step was to explore whether meaningful and significant coherences could be found in the time-frequency space
between the equally spaced and 7.5yr lowpassed speleothem $\mathit{\Delta}^{13}C_{Baradla}$ time series and similarly lowpassed primary
climate parameters (precipitation, temperature). In the case of the composite $\mathit{\Delta}^{13}C_{Baradla}$ record, the best coherence was
found with the December-March average precipitation amount, and this was mostly in anti-phase.
The generally anti-phase coherence/relationship between precipitation amount and the composite carbon isotope record
is in accordance with the general theory that a higher amount of precipitation results in enhanced biological soil activity,
more biogenic $CO_2$ in the soil, and consequently more negative $\mathit{\Delta}^{13}C$ values in the speleothems of the Baradla Cave
(Demény et al., 2017). Although the relationship between the $\mathit{\Delta}^{13}C$ and precipitation amount had previously been observed
for individual records using visual comparisons and regression analysis (Demény et al., 2017), it was found to be weak
due to dating uncertainties and the varying effect of additional factors contributing to the precipitation-composition
relationship (e.g. kinetic fractionation, vegetation change, prior calcite precipitation; see (Fairchild and Baker, 2012) for
the compilation of governing factors). Nevertheless, with wavelet coherence analysis, it was not the whole spectra which
was taken into account all at once (as in the case of linear regression analysis), but the relationship was mapped for each
frequency band and observed over time. The phase of the coherence was observed to vary on the decadal-scale. A
generally negative phase difference was revealed between $\mathit{\Delta}^{13}C$ and precipitation amount (Fig. 3).

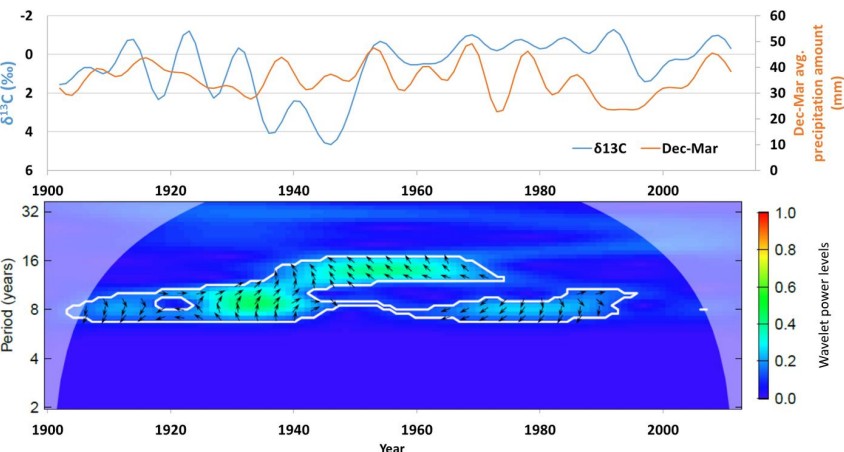


**Figure 3: Time series of the 7.5yr lowpassed composite $\mathit{\Delta}^{13}C_{Baradla}$ records (on a reversed axis) with the gridded**
**December-March average precipitation amount (Harris et al., 2014) (upper panel) and their time–frequency**
**coherency image (lower panel). The white contours in the lower panel show the 90% confidence levels calculated**
**based on 1000 AR (1) surrogates. The black arrows indicate the phase-angle difference.**






Significant coherences were found between the Baradla speleothem's $\Delta^{18}O_{Baradla}$ records and the monthly- and multi-
monthly averages of the primary climate parameters with a common discontinuity around the 1960s (Fig. 4). Such
coherences were to be expected, since the oxygen isotope composition of the speleothem is driven by temperature and
drip water composition (Lachniet, 2009), with the latter directly related to meteoric water composition governed by
atmospheric temperature and moisture origin (Dansgaard, 1964; Kaiser et al., 2001).
However, with regard to the phase differences between the $\Delta^{18}O_{Baradla}$ records and the climate parameters, a somewhat
confusing picture can be observed in the investigated ~110 years. The phase differences changed multiple times, and a
dominant direction could hardly be assigned. This may be a result of the complex interplay of governing factors.

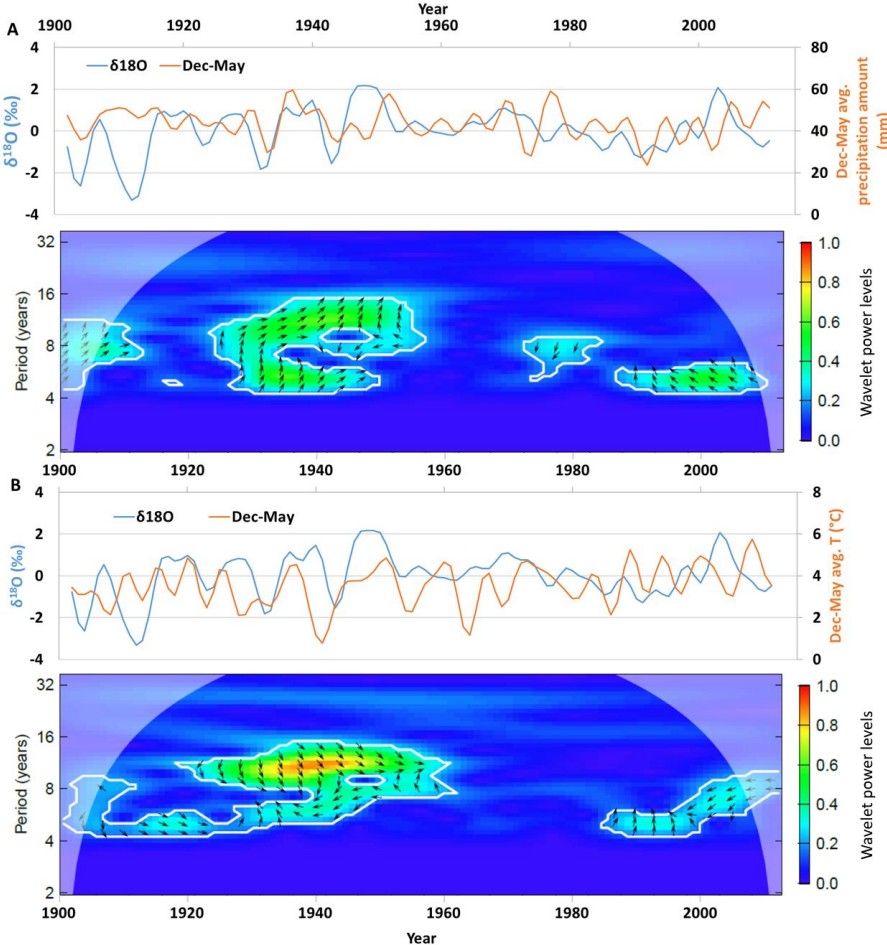


**Figure 4: Time series of the 4.5yr lowpassed composite $\Delta^{18}O$ records with the gridded climate data (Harris et al.,**
**2014) December-May average (a) temperature and (b) precipitation (upper panels) and their time–frequency**
**coherency image (lower panels). For further details see the caption of Fig. 3.**

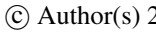


In order to extend the investigation on the climate-composition relationship through the exploration of the moisture origin
effect, the $\Delta^{18}O_{Baradla}$ record was compared with NAO indices that quantify the large-scale atmospheric processes directly
influencing moisture source pathways over the region of interest. Out of the multi-monthly averages, the strongest
coherence was found with the December-May averages of the station-based (Fig. 5a) and PC-based (Fig. 5b) NAO
records. The power maxima of the significant coherences (> 0.25) were reached at the ~ 8 yr period, and the relationship
was dominantly in anti-phase. Moreover, in a strong negative NAO mode, their coherence with the speleothem $\Delta^{18}O$
record was disrupted (Fig. 5).

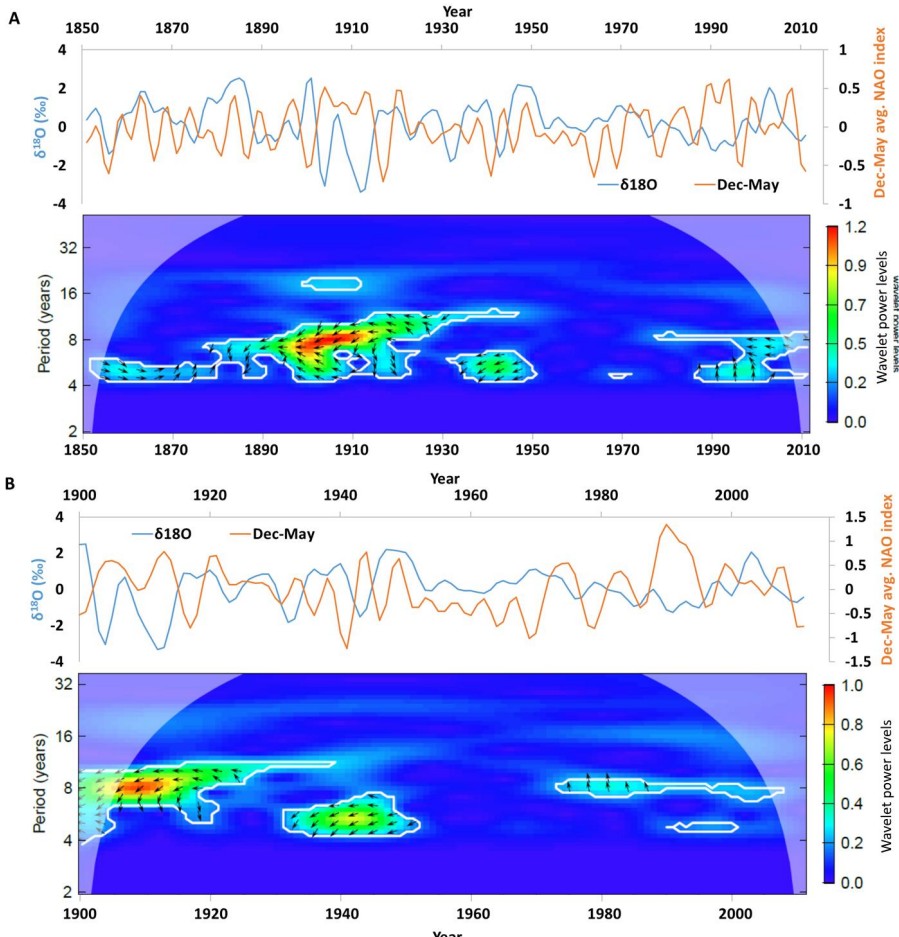


**Figure 5: Time series of the 4.5yr lowpassed composite $\Delta^{18}O$ records with the December-May average (a) NAO$_{st}$**
**and (b) NAO$_{PC}$ reconstructions (upper panels) and their time–frequency coherency images (lower panels). For**
**further details see the caption of Fig. 3.**



The generally anti-phase relationship with the NAO indices suggests that the $\Delta^{18}$O values of Baradla stalagmites mainly
reflect variations in moisture origin and, hence, fluctuations in moisture transport trajectories. In a negative NAO mode
the dominance of moisture transport is shifted to the Mediterranean, from where more $^{18}$O-rich moisture arrives to the
Eastern Alps (Kaiser et al., 2001), and also to the Baradla Cave region, in accordance with the anti-phase relationship
observed in this study. This observation can be used to interpret further the isotope data of older speleothems in the light
of the large scale atmospheric and oceanic variations of the North Atlantic realm.

## 4. Speleothem stable isotope records in a strongly NAO-influenced area

The methodology proposed and utilized on the Baradla speleothem was tested on an additional high resolution and
accurately dated speleothem dataset. The Proserpine stalagmite's $\delta^{13}$C and $\delta^{18}$O records (Han-sur-Lesse Cave, Belgium;
Van Rampelbergh et al., 2015) covering the period 1479-2000 AD were processed following the same procedure.
Although in the text only the section comparable with the previously discussed $NAO_i$ spanning 1857-2000 is processed
and compared using WTC with the $NAO_i$, the processed datasets for the entire time can be found in the supplementary
material. After the necessary processing to achieve the even spacing of the records, the spectral assessment of the rLSPs
of the original and the interpolated data indicated that a 3.5yr and 3.3yr lowpass filtering is necessary for the full $\delta^{18}$O
and $\delta^{13}$C records of the Proserpine speleothem, respectively, to avoid bias caused by the interpolation. As a next step the
season with strongest response was selected, by forming multi-monthly averages of the climate data, as in the case of the
Baradla speleothem stable isotope records. Note here that the climate data was lowpassed, just as the speleothem data
assessed together with it.
The relationship with the precipitation reconstruction was explored, and it proved to be mostly in-phase (where it gave
any relationship at all, only from ~1960 onwards), most probably due to the superimposed effects of local factors and
regional climate variations. With the $NAO_i$, however, it indicated changing (Fig. 6a) and an anti-phase coherency (Fig.
6b) at the ~8yr period band.
The Proserpine $\delta^{18}$O record indicated a strengthened response with the November-March averages of the $NAO_{st}$ and the
November-April averages of the $NAO_{PC}$ indices at the ~7yr period, predominantly in anti-phase (Fig. 7). The anti-phase
relationship was most explicit in the case of the $NAO_{PC}$ index (Fig. 7b). The strength of the coherence was somewhere
lower for the shorter comparison with the $NAO_{PC}$ than for the $NAO_{st}$ (Fig. 7) record, which had an extra ~50yrs overlap
with the speleothem stable isotope record.
The reason why the NAO and the Proserpine $\delta$ 18O records are in anti-phase is because when the NAO index is positive,
the cave receives more winter (low- $\delta^{18}$O) precipitation, while the negative NAO state induces cold and dry periods
around the cave environment, resulting in increased $\delta^{18}$O values in the speleothem (Van Rampelbergh et al., 2015). The
anti-phase $\delta^{18}$O-$NAO_i$ relationship is especially significant for the PC-based $NAO_i$. The detection of the relationship
between the processed Proserpine speleothem record (Van Rampelbergh et al., 2015) and the $NAO_i$ further validates the
procedure elaborated in this study and demonstrates the use of the established algorithm.





**Figure 6: Time series of the 3.3yr lowpassed composite $\Delta^{13}C$ records with the December-May average (a) NAO$_{st}$**
**and (b) NAO$_{PC}$ reconstructions and their time–frequency coherency image. For further details see the caption of**

291                      **Fig. 3.**


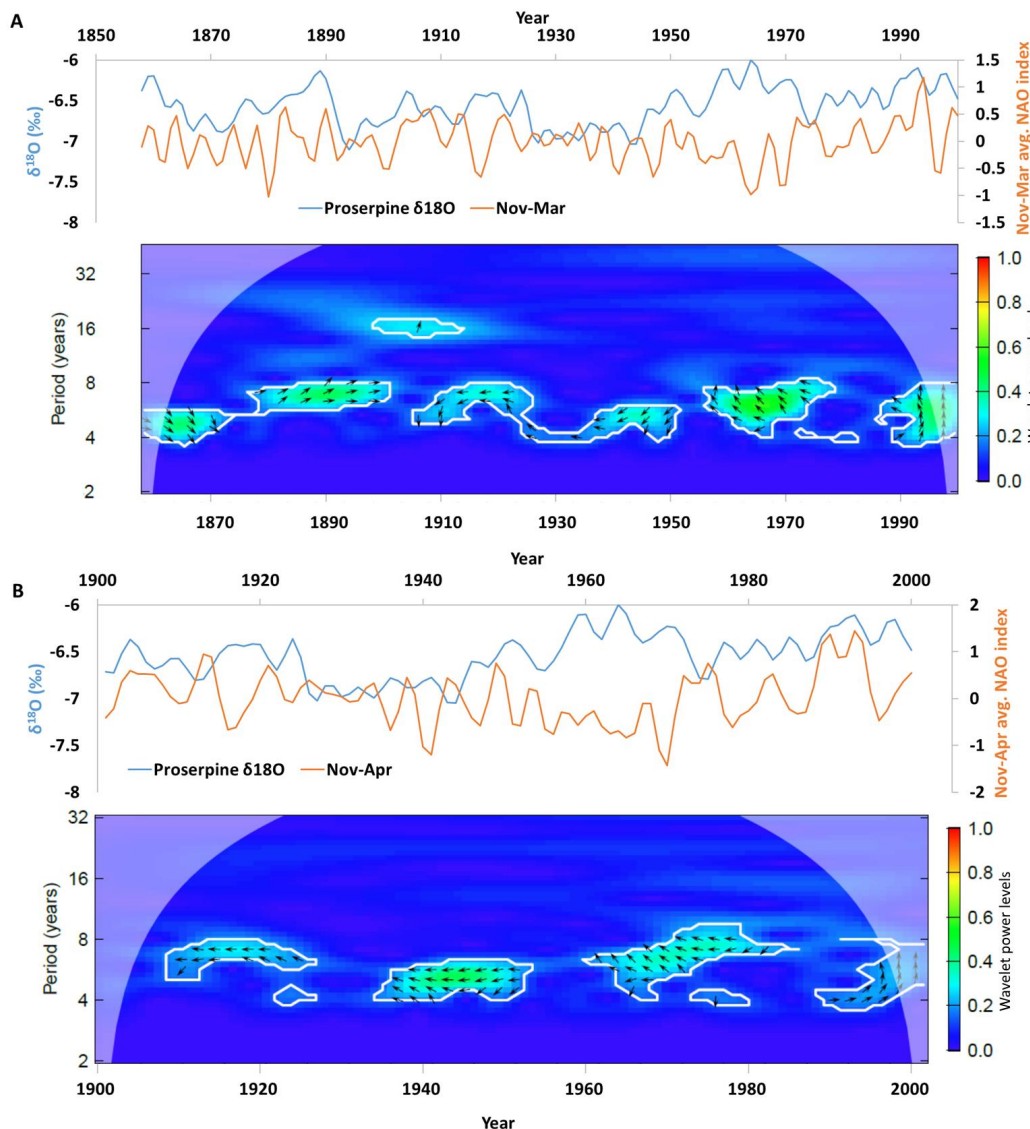


**Figure 7: Time series of the 3.5yr lowpassed Proserpine $\Delta^{18}O$ records with the (a) November-March average**
**NAO$_{st}$ and (b) December-April average NAO$_{PC}$ reconstructions (upper panels) and their time–frequency**
**coherency image (lower panels). For further details see the caption of Fig. 3.**

**5.  Conclusions**
A cubic spline-based universally applicable methodology was developed to handle the uneven spacing of sedimentary
proxy records, additionally taking into account the bias that interpolation may cause. The methodology was successfully
applied to the composite stable carbon and oxygen isotope records of the speleothems of the Baradla Cave, NE Hungary
and in addition to the similar records of a speleothem from the Han-sur-Lesse Cave, Belgium.



The composite stable isotope records of the Baradla Cave speleothem were compared with monthly-resolved temperature
and precipitation-amount data using wavelet transform coherence analyses. A generally negative relationship between the
carbon isotope record and cold season precipitation amount was revealed, in accordance with earlier assumptions. In the
case of the oxygen isotopic composition, its relationship with primary climate variables (temperature and precipitation
amount) was less clear, probably due to the competing and/or superimposed factors that determine the carbonate oxygen
isotopic composition. Nevertheless, the $\Delta^{18}O$ record indicated an anti-phase relationship with North Atlantic Oscillation
indices reflecting moisture origin. Specifically, in a negative NAO mode the moisture transport trajectory is shifted to the
Mediterranean from where the Baradla Cave receives $\delta^{18}O$-enriched precipitation. These observations provide a firm base
for the interpretation of stable isotope data obtained for the Baradla Cave system, NE Hungary. The now evenly resampled
and lowpass filtered composite records can serve as regional benchmarks in future proxy paleoclimate evaluations.
Moreover, the methodology was successfully applied using published data for a stalagmite from Belgium ($\delta^{13}C$ and $\delta^{18}O$
records of the Proserpine stalagmite, Han-sur-Lesse Cave). Expressed relationships with the NAO$_i$, especially with the $\delta$
$^{18}O$ records, were found in accordance with the earlier assumptions, lending credibility to the methodology applied and
developed in this study.

**6. Author contribution**
I.G. Hatvani developed the model code and performed the simulations. Z. Kern preprocessed the data and provided the
composite records. A. Demény conceived the study and provided the geochemical interpretation. Sz. Leél-Őssy was
responsible for the cave studies. All authors took part in the manuscript preparation.

**7. Data availability:**
The data sets produced with the methodology presented in the paper are available at doi: 10.1594/PANGAEA.875917.

**8. Competing interests:**
The authors declare that they have no conflict of interest.

**9. Acknowledgements**
Financial support was received from the Hungarian Academy of Sciences (MTA "Lendület" program; LP2012-27/2012),
the National Research, Development and Innovation Office (OTKA NK 101664) and the János Bolyai Research
Scholarship of the Hungarian Academy of Sciences. This is contribution No. XX of 2ka Palæoclimatology Research
Group.

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
