# Peer review of "Speleothem stable isotope reference records for East-Central"

_Earth System Science Data, 2017_

## Short Comment (SC1) · 9 Aug 2017

As to the methodology, I think it would suffice if the stable carbon and oxygen isotope records derived from two stalagmites of the Baradla Cave(NE Hungary) dating back to the late 18th century and stable isotopes records from the Han-sur-Lesse Cave (Belgium) are employed. Well-grounded and well-written.

I found some related knowledge on this page: http://www.bocsci.com/isotope-labeling-service.html

---

## Referee Comment (RC1) · Anonymous Referee #1 · 30 Aug 2017

Dear Editor,

The manuscript 'Speleothem stable isotope reference records for East-Central Europe - Resampling sedimentary proxy records to get evenly spaced time-series with spectral control' by Hatvani et al. is pointing at an important aspect of palaeoclimate time series, namely, statistical analyses of un-evenly spaced time series. However, these statistical analyses often require evenly spaced time series. Therefore, in case of un-evenly palaeoclimate time series, various approached are used to equalize the temporal resolution of the palaeoclimate time series.

[Figure]

This manuscript illustrates on the base of published modern speleothems proxy (d18O and d13C) time series from two stalagmites how un-evenly spaced time series can be made evenly spaced to (in their case) an annual resolution using a cubic spline interpolation. Furthermore, the authors discuss the difference between the frequency spectrum (derived by REDFIT[1] and REDFIT-X[2]) of the un-evenly and evenly spaced speleothem proxy time series. From this comparison they derive a 'cut-off' period, the period at which the frequency spectrum of the un-evenly and evenly spaced speleothem proxy time series differs, and use this period to low-pass filter the (newly generated) evenly spaced speleothem proxy time series. This low-pass filtered evenly spaced speleothem proxy time series is than compared to modern meteorological time series such as temperature, precipitation and the index of the North Atlantic Oscillation using Wavelet transform coherency (WTC). Based on these WTC analysis they conclude that a NAO signature is imprinted in the speleothem proxy time series.

Although this study tackles a timely technical problem of palaeoclimate time series, the manuscript has several shortcomings, detailed listed in my general comments. Noteworthy is that similar ideas have been used/developed in other studies and this manuscript should state this. Before these general comments are not addressed I cannot recommend the publication of this manuscript. I also listed detailed comments after my general concerns.

General comments:

As stated in my comment to the editor, I think that this study is very timely, however, I miss a more general review of the palaeoclimate literature on this topic. What has been done so far and how does this study improve our knowledge compared to these studies. This includes on the one-hand palaeoclimate studies that have applied such approaches and on the other hand studies that discuss different smoothing/interpolation techniques. This speleothem-based study [3] is an example. They have used a very similar technique and approach as presented in this study, though there are some minor differences but the idea behind is the very same as for this study. But there are also

other techniques for sedimentary archives, such as from [4]. Furthermore, the authors should provide some more backups for using a cubic spline to fill gabs in an un-evenly spaced time series (e.g. [5] [6]).

What is the rational to use a cubic spline (or any other technique) for increasing (gab filling) the temporal resolution of time series, i.e., generation information? For the illustrated example, the used approach is probably fine for the youngest part of the speleothem proxy time series, with a very high temporal resolution, but the oldest part of the speleothem proxy time series, is much smaller compared to the youngest part. However, what is strikingly illustrated in Figure 1 of this manuscript is that the variability of younger part is much higher compared to the older part. Therefore, don't you assume that the higher variability in the younger part is noise? I think this approach needs a more rigorous test, whether it is adequately working or not by testing it with artificial time series using e.g. an AR-1 or AR-2 model. My suggestion is to generate an equally spaced time series (with known spectrum), which is than un-equally resampled n-times. Use these n un-equally spaced artificial time series to test your approach. Ideally you use different kinds of artificial time series to test your approach, e.g. AR-1, AR-2. For the WTC analysis you may consider to use an artificial time series, which is forced by the observed NAO index – add some noise or regional factors such as temperature – and test your approach again, this time including the WTC. I think these steps are necessary to really quantify the quality of your approach, but also to add new results to this topic (see comment on literature).

Comments:

Line 33: Please change the statement on the dating. The advantage of speleothems is, that they can be precisely dated by speleothems. My suggestion is: they can be precisely dated by U-series techniques (e.g. U-Th) Line 34: I would suggest rewriting part on multiple proxies. My suggestion is: they provide multiple independent proxy that are suitable to reconstruct past climate conditions. Line 36: Fairchild and Baker is not review article but a book. My suggestion is to change review to overview. Line

37-39: Please rephrase and split this sentence; it is quite complex and not very reader friendly. What does 'composition' mean here? I would also suggest to use 'determine' instead of 'govern'. Line 39-40: I think you have to distinguish between different proxies in this sentence. From what we know about proxies some are really modified by global processes (e.g. d18O) but others are clearly locally controlled, such as trace elements (TEs). Line 44: I would change 'behaviour' to 'dynamics' or 'climate'. Line 44-45: You don't really get information on speleothem formation processes by cave monitoring but information on speleothem formation. I would suggest to delete processes here. Line 46-48: I think you have to be more specific here, when you refer to seasonality. Readers may not be familiar with cave monitoring and to cave parameter to which you refer seasonality (I think). E.g. seasonality of cave air temperature or wind. Line 56-58: You need to rephrase and improve the English of his sentence. Be more specific on the moving average and what it means for TIME series. Line 60: Delete the 'and' between 'statistical' AND 'time series', because it does not make sense at the moment. You refer to statistical time series analysis. Line 66: I think you should add that these 'common' statistical tools can only be used for evenly spaced time series. Line 66: You should add 'unevenly spaced' before sedimentary records. Line 67: There is 'by' missing. BY using. Line 67: include 'techniques' after non-linear. Line 68: What are 'insufficiently documented methods'? Can you be more specific? Line 72: Where is Baradla Cave located? Line 71-73: This is a very long sentence and I would suggest to split and rephrase it. Line 75: include 'packages' after software. Line 78-79: This last sentence makes no sense in the context of the previous sentence. Line 83: I think it is helpful for the reader to give a short rationale why these two stalagmites were used compared to the other two of the same caves, e.g., from the summary/conclusion of the original study [7]. Line 87-90: How are the age models constructed of these two stalagmites? How large are the age uncertainties? Is it reasonable, based on the age uncertainties' to merge them? Line 90: I think you should use a different symbol for the merged record than $\Delta$, which is used for clumped isotopes and for differences between time series for example. It is okay to use $\delta$ here. Line 93: You can delete 'totals'. Line 152-

153: Why do you low-pass filter your data? This is in principle not necessary if you use WTC, because you can neglect any observed coherence (if any) below your threshold frequency. In this case you do not alter the meteorological observations and your interpolated time series and reduces further uncertainties. Line 165: I suppose you mean "In THIS study" not "In the study". Line 167-168: The fact that other speleothem studies have used the Morlet mother wavelet is no argument. I suggest to delete this part of the sentence. Line 171: What is a "positive signal"? I suggest to rephrase this sentence; be more specific.

Line 174-176: This is a very nice approach, but it needs a more detailed explanation. Maybe show some results and draw the steps you perform until you find the final result. Line 184-186: How about the small periods? There is quite a difference between the original and interpolated time series between 32 and 64 years. Line 188: What is the rational to determine the threshold exactly at 4.5 years? Line 197-198: What is the rational to determine the threshold exactly at 7.5 years? Line 203-205: See my comment on line 152-153. Line 226: What are the primary climate parameters? Although it is stated in Figure 4, please include it in this sentence. Why not write "precipitation and temperature" instead of "primary climate parameters"? Line 226-229: Apart from Kaiser (2001), Lachniet (2009) and Dansgaard (1964) generally explain global phenomena. Is this also the case for local precipitation d18O values? Line 239-241: Are there references for this statement? Line 241-243: What is your definition of "strong"? Does it refer to the Wavelet power or the length of the period during which you observe a significant coherence. However, the coherence between the two NAOi and the interpolated d18O time series is low apart from a 10 year long period in the beginning of the 20th century. Line 244-245: Please state clearly which years you mean or identify these years in Figure 5. Line 252-253: Please state a reference for this or cite Kaiser earlier in the paper. Line 266: Can you show the results of the redfit-x analyses in an additional figure in the supplementary information. Line 269-270: See my comments on low-pass filtering. Line 271: The relationship of what? d18O or d13C? Where are the results of this comparison?

Figure captions:

Figure 3: Please include a detailed description of the arrows. When is the signal in-phase, pointing right? Left? Figure 5: Can you rescale the time axis (figure) of panel b that the time on panel a and b can be compared. This makes it easier for reader to compare the coherence between the two different NAOi and the interpolated d18O time series.

References: 1. Schulz, M. and M. Mudelsee, REDFIT: Estimating red-noise spectra directly from unevenly spaced paleoclimatic time series. Computers & Geosciences, 2002. 28(3): p. 421-426. 2. Ólafsdóttir, K.B., M. Schulz, and M. Mudelsee, REDFIT-X: Cross-spectral analysis of unevenly spaced paleoclimate time series. Computers & Geosciences, 2016. 91: p. 11-18. 3. Cosford, J., et al., East Asian monsoon variability since the Mid-Holocene recorded in a high-resolution, absolute-dated aragonite speleothem from eastern China. Earth and Planetary Science Letters, 2008. 275(3): p. 296-307. 4. Lisiecki, L.E. and P.A. Lisiecki, Application of dynamic programming to the correlation of paleoclimate records. Paleoceanography, 2002. 17(4). 5. Musial, J.P., M.M. Verstraete, and N. Gobron, Comparing the effectiveness of recent algorithms to fill and smooth incomplete and noisy time series. Atmospheric chemistry and physics, 2011. 11(15): p. 7905-7923. 6. Hocke, K. and N. Kämpfer, Gap filling and noise reduction of unevenly sampled data by means of the Lomb-Scargle periodogram. Atmospheric Chemistry and Physics, 2009. 9(12): p. 4197-4206. 7. Demény, A., et al., Recently forming stalagmites from the Baradla Cave and their suitability assessment for climate–proxy relationships. Central European Geology, 2017. 60(1): p. 1-34.

---

## Referee Comment (RC2) · L. Comas-Bru (Referee) · 8 Sep 2017

This manuscript addresses an important issue with paleoclimate records, namely that the time interval between samples is not regular. This issue has been addressed in previous studies such as Schulz et al 1997 (Computers & Geosciences 23 (9), 929–945), Schulz and Mudelsee 2002 (Computers & Geosciences 28, 421–426), Björg Ólafsdóttir et al 2016 (doi: 10.1016/j.cageo.2016.03.001) and Rehfeld et al., 2011 (doi: 10.5194/npg-18-389-2011). In the first three, SPECTRUM, REDFIT and REDFIT-X are used to calculate the spectra and red-noise of unevenly spaced samples while the

fourth provides a thorough comparison of different correlation analysis techniques for irregularly spaced samples.

This study uses two published records of oxygen and carbon isotopes from two speleothems from Hungary and Belgium to illustrate this issue. The authors transform these unevenly spaced series to a temporally equidistant time series suitable for the application of common statistical methods such as correlation techniques. In particular, they compare their reconstructed time series to low-pass filtered meteorological data and the NAO index and conclude that the signal of the latter is recorded in the speleothem time series for certain band-widths.

General comments

How to deal with uneven time-series is an important issue in paleoclimate which has been previously addressed. However, the authors fail to do a more extensive comparison between their approach and other already used techniques.

The length of the manuscript used to interpret what were supposed to be only examples of the technique is, from my point of view, losing the focus of the main objective: providing a technique to deal with uneven time-series (see main aim of the paper in L71-73).

The authors are omitting some fundamental issues of speleothem growth by not including a proper discussion on how the water storage time in the karst may yield an auto-correlated isotopic time-series. This becomes important when interpreting the speleo record in terms of climate variables if the storage component in the aquifer is larger than the cut-off period numerically chosen from the Lomb-Scargle Fourier Transform periodogram.

In addition to this, interpreting paleoclimate records only in terms of spectra analysis is complex if the physical mechanisms at such periodicities is not properly reviewed. For example, how do the authors know that the anti-phase coherence shown in Figure 3

(top panel) for a specific frequency band is related to biological soil activity (L207-209) and not to another controlling factor? Similarly, how do the authors know that changes in the moisture source are the dominant controlling factor in that specific frequency band, as suggested in L252-253? More discussion on this would make the whole Section 3 more robust.

The authors do not use single speleothem records but a composite record. When using composites, the temporal and proxy uncertainties increase, but these are not being dealt with in the manuscript. In addition, isotopic records from individual speleothems from the same cave may vary quite a lot depending on the groundwater history within the aquifer that eventually reaches each of the drip sites. I assume that this was mentioned in previous research but I suggest that if the authors want to use this composite record more information is provided. However, my suggestion would be to analyse both records separately and then compare it to the composite. This alternative approach would highlight potential discrepancies between the records and would make the study more robust.

I wonder if the composite uncertainties have an effect on the periodogram used to decide the appropriate cut-off period if they are considered with an iterative approach. Do significant peaks remain significant? How do these uncertainties affect the spectra analysis of the records when it comes to interpreting in terms of climate variables? This could be also argued for a single speleothem record, as all proxy records have dating and measurement uncertainties.

I do not see any benefit in including the speleothems from Belgium in this study. They are just another example and they're only mentioned in the 25 lines of the discussion section, just before the conclusions. I suggest the authors either omit these records or discuss them more extensively.

Minor comments

L33: Aren't all dating methods "numerical"?

L34: The temporal resolution of the speleothem proxy record will depend on the karst processes at play at that particular drip site. This can hugely vary from sub-annual to centennial resolution and therefore, it is not always possible to obtain a highly resolved temporal record. Regarding the "high spatial resolution"? I'd suggest that "they are well distributed worldwide" instead. Would it be possible to know what are the benefits of using a cubic spline in contrast to other interpolation methods?

L52: I don't understand the concept of "reference records". Why are these records considered to be a reference for their regions? This manuscript does not compare several series from the same region, so I think the usage of this term can become misleading.

L52-53: An additional criteria to select "reference records" according to the authors, should be the overlap with meteo data (you could find accurately dated records close to annual that do not overlap with meteorological instrumental data).

L86: There is no need to include the definition of delta here.

L92: "see section 2.1". That sentence is already part of section 2.1!

L106-109: It doesn't really matter for the purpose of this manuscript but you can obtain a longer PC-based NAO index from the 20CRv2c dataset, which goes back to 1851 (https://www.esrl.noaa.gov/psd/data/20thC_Rean/)

L117-118: I would like to see the periodogram of the original record along with the one from the reconstructed signal early in the text. In contrast, Figure 1 does not provide much information (it is difficult to visually compare both panels). This is related to my comment below for L134.

L244-245: Which period corresponds to a "strong negative NAO mode" and for which frequency-band is this evident?

L134: I would like to see a figure showing the significant powers of bot series (original and reconstructed)

The use of "composition" when referring to a composite series is confusing (for example "precipitation-composition" relationship in L212. Also, in L37, the usage of "composition" seems to mean "proxy records". Please clarify.

L226: "...with the primary climate parameters", these being?

Fig 5 (and others): could you please mark the cut-off periods below which the spectral analyses are not significant?

---

## Author Comment (AC1) · 8 Sep 2017

We would like to thank the Reviewer #1 for her/his suggestions. We would like to give an update on the progress of the test she/he suggested regarding, what we think is the most crucial issue in her/his comments.

Based on the suggestion, we selected a well-studied record with a pronounced periodic signals to test the performance of our approach. Seasonal averages were computed from the monthly mean total sunspot number (WDC_SILSO, 2017) and randomly re-

sampled. Seven data points were taken randomly from each block of 10yrs to replicate the sub-annual resolution of the Baradla speleothem (avg. 0.7 stable isotope data/yr). This was repeated a 100 times, so an ensemble 100 randomly resampled time series were obtained which replicate the resolution of the Baradla $\delta$18O records. As a final step in line with the proposed protocol the 100 randomly resampled time series were spline interpolated and annual averages were formed and were processed using RED-FIT to assess their spectral characteristics and compare them to the spectra of the annual averages of the original (sunspot) record.

All the 100 redfit Lomb-Scargle Periodograms (rLSPs) replicated the well-known ~11 yr sunspot cycle ($\alpha$>0.95). Moreover, most of the rLSPs of the un-equally spaced artificial time series mirrored even the smaller peaks (e.g. ~8yrs). However, the noise caused by spline interpolation surfaced in the high-frequency domain (Fig. 1). In our view, these results provide a proper support for the yet untested assumption in the MS.

We hope these results back up the applicability of the methodology proposed on sedimentary proxy datasets, as the ones in the MS. The following results are only preliminary and intend to show the Reviewer #1 the promising progress of the rigorous test she/he suggested. If approved by the Reviewer #1, the results of this test is planned to be attached to the revised version of the MS in a form of a supplement in a more detailed way. Data source: WDC-SILSO, Solar Influences Data Analysis Center (SIDC), Royal Observatory of Belgium, Av. Circulaire, 3, B-1180 BRUSSELS Currently at http://www.sidc.be/silso/datafiles accessed on 08.09.2017

Fig. 1. rLSP of the annual mean sunspot numbers (red); and rLSPs of the 100 un-equally spaced artificial (sunspot) time series randomly resampled, spline interpolated and annualized from the monthly mean sunspot numbers (black). The bias-corrected 95% chi-squared limit of a fitted AR(1) process for the rLSPs is shown in orange.

[Figure]

**Fig. 1.** (please see figure caption above)

---

## Author Comment (AC2) · 8 Sep 2017

In the lack of any any specific suggestion, we would only like to thank you for your comment.

---

## Author Comment (AC3) · 9 Oct 2017

**The answers to the questions of Reviewer #1**

**General comments**
**As stated in my comment to the editor, I think that this study is very timely, however, I miss a more general review of the palaeoclimate literature on this topic. What has been done so far and how does this study improve our knowledge compared to these studies. This includes on the one-hand palaeoclimate studies that have applied such approaches and on the other hand studies that discuss different smoothing/interpolation techniques. This speleothem-based study [3] is an example. They have used a very similar technique and approach as presented in this study, though there are some minor differences but the idea behind is the very same as for this study. But there are also other techniques for sedimentary archives, such as from [4].**

Thank you for the suggestion, the paper of Cosford et al. (2008) was indeed relevant from multiple aspects. Therefore, besides extending the introduction with other examples, Cosford et al. (2008) has been incorporated in lines 64-69 as: "A more complex approach combining spline smoothing and linear interpolation was presented in a multi paleoproxy study, where in addition the spectral characteristics of the data were assessed using multiple methods as well (Cosford et al., 2008). However, any transformation which adds data to or removes it from the original record unavoidably changes its spectral characteristics. This is a factor, which can easily be overlooked even in advanced studies although it must deserve high attention."
In the last sentence we implicitly refer to the study of Cosford et al. (2008) who only used the Nyquist frequency of the data to determine a cut-off period. On the contrary, the procedure in the present paper considers the spectrum specific characteristic when determining a cut-off frequency.

Unfortunately, the study of Lisiecki (2002) mainly deals with signal matching and even states that hiatuses may produce erroneous results in his approach, thus we would not include this paper in the reference list.

Reference added:

Cosford, J., Qing, H., Eglington, B., Mattey, D., Yuan, D., Zhang, M., and Cheng, H.: East Asian monsoon variability since the Mid-Holocene recorded in a high-resolution, absolute-dated aragonite speleothem from eastern China, Earth and Planetary Science Letters, 275, 296-307.

**Furthermore, the authors should provide some more backups for using a cubic spline to fill gabs in an un-evenly spaced time series (e.g. [5] [6]). What is the rational to use a cubic spline (or any other technique) for increasing (gab filling) the temporal resolution of time series, i.e., generation information?**

Thank you for the comment, the papers below have been incorporated. Thus, the text regarding the benefits and drawbacks of spline interpolation has been thoroughly extended and it is now briefly compared to linear interpolation, the Lomb-Scargle Reconstruction method and the Kondrashov and Ghil technique.

References added:

Musial, J.P., M.M. Verstraete, and N. Gobron, Comparing the effectiveness of recent algorithms to fill and smooth incomplete and noisy time series. Atmospheric chemistry and physics, 2011. 11(15): p. 7905-7923.

Hocke, K. and N. Kämpfer, Gap filling and noise reduction of unevenly sampled data by means of the Lomb-Scargle periodogram. Atmospheric Chemistry and Physics, 2009. 9(12): p. 4197-4206.

Kondrashov, D. and Ghil, M.: Spatio-temporal filling of missing points in geophysical data sets, Nonlin. Processes Geophys., 13, 151-159, doi: 10.5194/npg-13-151-2006, 2006.

**For the illustrated example, the used approach is probably fine for the youngest part of the speleothem proxy time series, with a very high temporal resolution, but the oldest part of the speleothem proxy time series, is much smaller compared to the youngest part. However, what is strikingly illustrated in Figure 1 of this manuscript is that the variability of younger part is much higher compared to the older part. Therefore, don't you assume that the higher variability in the younger part is noise?**

The main aim was to create an equally sampled dataset which has/resembles the spectral characteristics of the original data. During the procedure, the signal to noise ratio of the original data is not assessed, only the bias (introduced artificial noise) caused by the interpolation itself as shown in Section 2.2.1.

**I think this approach needs a more rigorous test, whether it is adequately working or not by testing it with artificial time series using e.g. an AR-1 or AR-2 model.**

**My suggestion is to generate an equally spaced time series (with known spectrum), which is than un-equally resampled n-times. Use these n un-equally spaced artificial time series to test your approach. Ideally you use different kinds of artificial time series to test your approach, e.g. AR-1, AR-2. For the WTC analysis you may consider to use an artificial time series, which is forced by the observed NAO index – add some noise or regional factors such as temperature – and test your approach again, this time including the WTC. I think these steps are necessary to really quantify the quality of your approach, but also to add new results to this topic (see comment on literature).**

Based on the Reviewer's suggestion, we selected a well-studied record with pronounced periodic signals to test the performance of our approach. Seasonal averages were computed

from the monthly mean total sunspot number (WDC-SILSO, 2017) and randomly resampled. Seven data points were taken randomly from each block of 10yrs to replicate the sub-annual resolution of the Baradla speleothem (avg. 0.7 stable isotope data/yr). This was repeated a 100 times, so an ensemble of 100 randomly resampled time series were obtained which replicate the resolution of the Baradla $\delta^{18}O$ records. As a final step in line with the proposed protocol the 100 randomly resampled time series were spline interpolated and annual averages were formed and were processed using REDFIT to assess their spectral characteristics and compare them to the spectra of the annual averages of the original (sunspot) record.

All the 100 redfit Lomb-Scargle Periodigrams (rLSPs) replicated the well-known ~11 yr sunspot cycle ($\alpha>0.95$). Moreover, most of the rLSPs of the un-equally spaced artificial time series mirrored even the smaller peaks (e.g. ~8yrs). However, the noise caused by spline interpolation surfaced in the high-frequency domain (Fig. R1).

[Figure]

**Fig. R1. rLSP of the annual mean sunspot numbers (red); and rLSPs of the 100 un-equally spaced artificial (sunspot) time series randomly resampled, spline interpolated and annualized from the monthly mean sunspot numbers (black). The bias-corrected 95% chi-squared limit of a fitted AR(1) process for the rLSPs is shown in orange.**

The results shown above have been added to the MS in lines 166-170 and in a detailed form in Section S1 of the supplementary online material.

Data source:
WDC-SILSO, Solar Influences Data Analysis Center (SIDC), Royal Observatory of Belgium, Av. Circulaire, 3, B-1180 BRUSSELS Currently at http://www.sidc.be/silso/datafiles accessed on 08.09.2017

**Comments:**

**Line 33: Please change the statement on the dating. The advantage of speleothems is, that they can be precisely dated by speleothems. My suggestion is: they can be precisely dated by U-series techniques (e.g. U-Th)**

The sentence in question has been rephrased in line 30.

**Line 34: I would suggest rewriting part on multiple proxies. My suggestion is: they provide multiple independent proxy that are suitable to reconstruct past climate conditions.**

Accepted and corrected in lines 28-33.

**Line 36: Fairchild and Baker is not review article but a book. My suggestion is to change review to overview.**

Accepted and corrected

**Line 37-39: Please rephrase and split this sentence; it is quite complex and not very reader friendly. What does 'composition' mean here? I would also suggest to use 'determine' instead of 'govern'.**

The confusing sentence has been removed.

**Line 39-40: I think you have to distinguish between different proxies in this sentence. From what we know about proxies some are really modified by global processes (e.g. d18O) but others are clearly locally controlled, such as trace elements (TEs).**

Accepted and corrected. Your suggestion has been added to lines 35-36. as: "Although, some proxies are more likely to be influenced by large scale factors (e.g. oxygen stable isotope content), while others are clearly locally controlled (e.g. trace elements)."

**Line 44: I would change 'behaviour' to 'dynamics' or 'climate'.**

Accepted and corrected, behavior has been changed to dynamics.

**Line 44-45: You don't really get information on speleothem formation processes by cave monitoring but information on speleothem formation. I would suggest to delete processes here.**

Accepted and corrected, the word processes has been removed.

**Line 46-48: I think you have to be more specific here, when you refer to seasonality. Readers may not be familiar with cave monitoring and to cave parameter to which you refer seasonality (I think). E.g. seasonality of cave air temperature or wind.**

Accepted and corrected, the sentence has been extended in line 45.

**Line 56-58: You need to rephrase and improve the English of his sentence. Be more specific on the moving average and what it means for TIME series.**

The paragraph in question has been removed.

**Line 60: Delete the 'and' between 'statistical' AND 'time series', because it does not make sense at the moment. You refer to statistical time series analysis.**

The sentence has been rephrased in lines 54-55

**Line 66: I think you should add that these 'common' statistical tools can only be used for evenly spaced time series.**

Accepted and corrected in line 66

**Line 66: You should add 'unevenly spaced' before sedimentary records.**

Accepted and corrected.

**Line 67: There is 'by' missing. BY using.**

Accepted and corrected.

**Line 67: include 'techniques' after non-linear.**

Accepted and corrected.

**Line 68: What are 'insufficiently documented methods'? Can you be more specific?**

We would rather not extend this sentence in the MS. We referred to methodologies insufficiently documented in published papers to such an extent, that make reproducibility impossible.

**Line 72: Where is Baradla Cave located?**

Accepted and corrected, the sentence has been extended and a reference has been added to Section 2.1 in line 71 as: "… Baradla Cave (NE Hungary for details see Section 2.1)…".

**Line 71-73: This is a very long sentence and I would suggest to split and rephrase it.**

Accepted and corrected.

**Line 75: include 'packages' after software.**

Accepted and corrected.

**Line 78-79: This last sentence makes no sense in the context of the previous sentence.**

Accepted and corrected, the sentence has been removed.

**Line 83: I think it is helpful for the reader to give a short rationale why these two stalagmites were used compared to the other two of the same caves, e.g., from the summary/conclusion of the original study [7]. Line 87-90: How are the age models constructed of these two stalagmites? How large are the age uncertainties? Is it reasonable, based on the age uncertainties' to merge them?**

Accepted and corrected. Section 2.1 has been thoroughly extended with a discussion on the similarities of the two speleothems (NU2 & VK1) the composite was derived from, regarding the most important aspects: age model construction and synchronization; see lines (80-92). This was mainly extracted and summarized from Demény et al. (2017). Moreover, two new panels (A & B) have been added to Fig. 1 showing the complementary characteristics of the stable isotope records of the two speleothems.

**Line 90: I think you should use a different symbol for the merged record than $\Delta$, which is used for clumped isotopes and for differences between time series for example. It is okay to use here.**

Accepting the permission of the Reviewer, we keep the notation defined in lines 95-96, but we took special care to use $\Delta^{13}C_{Baradla}$ and $\Delta^{18}O_{Baradla}$ throughout the whole MS (including figures) to avoid any misunderstanding with $\delta^{13}C$ and $\delta^{18}O$ values. Moreover, the figure legends have been updated accordingly.

**Line 93: You can delete 'totals'.**

We would like to keep it.

**Line 152-153: Why do you low-pass filter your data? This is in principle not necessary if you use WTC, because you can neglect any observed coherence (if any) below your threshold frequency. In this case you do not alter the meteorological observations and your interpolated time series and reduces further uncertainties.**

The Reviewer is right, it can be a viable approach to neglect the frequency band below the chosen threshold frequency in a specific case of WTC. However, the idea here was to make the methodology universal and the last step was omitting the frequency bands that might have been biased by the interpolation. The main result is this output which can then be passed on to tools that require equally spaced data.

**Line 165: I suppose you mean "In THIS study" not "In the study".**

Accepted and corrected.

**Line 167-168: The fact that other speleothem studies have used the Morlet mother wavelet is no argument. I suggest to delete this part of the sentence.**

Accepted and corrected.

**Line 171: What is a "positive signal"? I suggest to rephrase this sentence; be more specific.**

Accepted and corrected, the term positive has been removed, it was indeed unnecessary.

**Line 174-176: This is a very nice approach, but it needs a more detailed explanation. Maybe show some results and draw the steps you perform until you find the final result.**

Accepted and corrected, a more detailed explanation was provided with exact examples in lines 198-204.

**Line 184-186: How about the small periods? There is quite a difference between the original and interpolated time series between 32 and 64 years.**

As stated in lines 170-173 the main idea was to match the peaks of the interpolated data's rLSP to the significant peaks of the original composite's rLSP. In this sense, the peak between the 32

and 64yr periods is insignificant not only in the original composite record's spectrum, but in the interpolated one as well at the 80% significance level. Please see Fig. R2 below.

[Figure]

**Fig. R2: rLSPs of the original, and the spline interpolated $\Delta^{18}O_{Baradla}$ records of the Baradla speleothem. The bias-corrected 80% chi-squared limit of a fitted AR(1) process for the raw composite's and the spline interpolated composite's rLSPs is shown in grey ad green respectively.**

**Line 188: What is the rational to determine the threshold exactly at 4.5 years?**

Accepted and the text has been corrected and the Reviewer is referred to lines 211-216 of the MS.

**Line 197-198: What is the rational to determine the threshold exactly at 7.5 years?**

Accepted and corrected, the text has been extended in lines 226-229.

**Line 203-205: See my comment on line 152-153.**

Please see our answer below your comment referring to lines 152-153.

**Line 226: What are the primary climate parameters? Although it is stated in Figure 4, please include it in this sentence. Why not write "precipitation and temperature" instead of "primary climate parameters"?**

Accepted and corrected, precipitation and temperature is now used in the sentence.

**Line 226-229: Apart from Kaiser (2001), Lachniet (2009) and Dansgaard (1964) generally explain global phenomena. Is this also the case for local precipitation d18O values?**

The text has been rephrased to be more specific, and an additional paper has been cited in lines 261-263 as: "Drip water composition is directly related to meteoric water composition governed by atmospheric temperature and moisture origin in the mid-latitudes (Dansgaard, 1964) and has been documented in the surroundings of the study area (Bottyán et al., 2017; Kaiser et al., 2001).

Reference added:

Bottyán, E., Czuppon, G., Weidinger, T., Haszpra, L., and Kármán, K.: Moisture source diagnostics and isotope characteristics for precipitation in east Hungary: implications for their relationship, Hydrological Sciences Journal, 62, 2049-2060,

**Line 239- 241: Are there references for this statement?**

The sentence in question has been shortened, the criticized part has been moved to Section 2.1. and extended and completed with supporting references in line 101 as: "Large scale tropospheric circulation controls precipitation amount and water stable isotope composition over Europe (Comas-Bru et al., 2016; Field, 2010)."

References added:

Comas-Bru, L., McDermott, F., and Werner, M.: The effect of the East Atlantic pattern on the precipitation $\delta$18O-NAO relationship in Europe, Clim Dyn, 47, 2059-2069, 2016.

Field, R. D.: Observed and modeled controls on precipitation $\delta$18O over Europe: From local temperature to the Northern Annular Mode, Journal of Geophysical Research: Atmospheres, 115, 2010.

**Line 241-243: What is your definition of "strong"? Does it refer to the Wavelet power or the length of the period during which you observe a significant coherence. However, the coherence between the two NAOi and the interpolated d18O time series is low apart from a 10 year long period in the beginning of the 20th century.**

The text has been extended with a reference to the wavelet powers. However, in the sentence we only stated that among all the available multi monthly averages, the one in Fig. 5 was found to be the strongest.

**Line 244-245: Please state clearly which years you mean or identify these years in Figure 5.**

Accepted and corrected in line 278 stating ~1955- ~1970.

**Line 252-253: Please state a reference for this or cite Kaiser earlier in the paper.**

The sentence in question has been removed from the text.

**Line 266: Can you show the results of the redfit-x analyses in an additional figure in the supplementary information. Line 269-270: See my comments on low-pass filtering. Line 271: The relationship of what? d18O or d13C? Where are the results of this comparison?**

Accepting the request of Reviewer #2 the corresponding Section 4, has been removed.

**Figure captions:**

**Figure 3: Please include a detailed description of the arrows. When is the signal inphase, pointing right? Left?**

Accepted and corrected.

**Figure 5: Can you rescale the time axis (figure) of panel b that the time on panel a and b can be compared. This makes it easier for reader to compare the coherence between the two different NAOi and the interpolated d18O time series.**

Accepted and corrected, please see Fig. 5 in the MS.

---

## Author Comment (AC4) · 9 Oct 2017

**The answers to the questions of Reviewer #2**

**General Comments**
**How to deal with uneven time-series is an important issue in paleoclimate which has been previously addressed. However, the authors fail to do a more extensive comparison between their approach and other already used techniques.**
Thank you for the comment. A highly relevant paper (Cosford et al., 2008) dealing with interpolation and spectral analysis of multiple E Asian stalagmites' $\delta^{18}O$ has been incorporated to lines 64-69 as: "A more complex approach combining spline smoothing and linear interpolation was presented in a multi paleoproxy study, where in addition the spectral characteristics of the data were assessed using multiple methods as well (Cosford et al., 2008). However, any transformation which adds data to or removes it from the original record unavoidably changes its spectral characteristics. This is a factor, which can easily be overlooked even in advanced studies, although, it must deserve high attention."
In the last sentence we implicitly refer to the study of Cosford et al. (2008) who only used the Nyquist frequency of the data to determine a cut-off period. On the contrary the procedure in the present paper considers the spectrum specific characteristic when determining a cut-off frequency.

Moreover, the text regarding the benefits and drawbacks of spline interpolation has been thoroughly extended as well and it is now briefly compared to linear interpolation, the Lomb-Scargle Reconstruction method and the Kondrashov and Ghil technique based on the following papers:

Musial, J.P., M.M. Verstraete, and N. Gobron, Comparing the effectiveness of recent algorithms to fill and smooth incomplete and noisy time series. Atmospheric chemistry and physics, 2011. 11(15): p. 7905-7923.
Hocke, K. and N. Kämpfer, Gap filling and noise reduction of unevenly sampled data by means of the Lomb-Scargle periodogram. Atmospheric Chemistry and Physics, 2009. 9(12): p. 4197-4206.
Kondrashov, D. and Ghil, M.: Spatio-temporal filling of missing points in geophysical data sets, Nonlin. Processes Geophys., 13, 151-159, doi: 10.5194/npg-13-151-2006, 2006.

**The length of the manuscript used to interpret what were supposed to be only examples of the technique is, from my point of view, losing the focus of the main objective: providing a technique to deal with uneven time-series (see main aim of the paper in L71-73).**
Accepted, thus Section 4 has been removed from the MS.

**The authors are omitting some fundamental issues of speleothem growth by not including a proper discussion on how the water storage time in the karst may yield an auto-correlated isotopic time-series. This becomes important when interpreting the speleo record in terms of climate variables if the storage component in the aquifer is larger than**

**the cut-off period numerically chosen from the Lomb-Scargle Fourier Transform periodogram.**

Thank you for the comment. The original signal should not be expected to be free from autocorrelation, which is amplified by water storage. It has been incorporated into Section 3.1. However, this does not question the introduction of the cut-off period to remove the artefact signals caused by interpolation techniques.

Moreover the introduction has been extended with a text referring to the infiltration dynamics of the Baradla Cave system above the samples in lines 86-92 as: "Thus, the best-match was obtained by shifting the NU2 and VK1 records by 3- and 5yrs respectively (Demény et al., 2017), supporting the synchronization of the individual isotope records. The good match of the isotopic peaks and the similar δ13C (Fig. 1a) and δ18O (Fig. 1b) amplitudes both argue for comparable water storage time in the feeding karstwater system. Based on lamina counting the chronological error at the base of VK1 is estimated to be between 3-5 yrs, however the perfect match (for δ13C see Fig. 1a and for δ18O see Fig. 1b;)) suggests the chronological difference to be much smaller. Moreover, the stable isotope records of the two stalagmites complement one-another by filling the hiatuses (Fig. 1a)."

**In addition to this, interpreting paleoclimate records only in terms of spectra analysis is complex if the physical mechanisms at such periodicities is not properly reviewed. For example, how do the authors know that the anti-phase coherence shown in Figure 3 (top panel) for a specific frequency band is related to biological soil activity (L207-209) and not to another controlling factor?**

Thank you for the suggestion, the sentence has been rephrased in line 239 and additional factors considered are discussed in the following sentences.

**Similarly, how do the authors know that changes in the moisture source are the dominant controlling factor in that specific frequency band, as suggested in L252-253? More discussion on this would make the whole Section 3 more robust.**

Thank you for the suggestion, the paragraph in question has been rephrased in lines 273-278.

**The authors do not use single speleothem records but a composite record. When using composites, the temporal and proxy uncertainties increase, but these are not being dealt with in the manuscript. In addition, isotopic records from individual speleothems from the same cave may vary quite a lot depending on the groundwater history within the aquifer that eventually reaches each of the drip sites. I assume that this was mentioned in previous research but I suggest that if the authors want to use this composite record more information is provided.**

Accepted and corrected. Section 2.1 has been thoroughly extended with a discussion on the similarities of the two speleothems (NU2 & VK1) the composite was derived from, regarding the most important aspects: age model construction and synchronization; see lines (80-92). This was mainly extracted and summarized from Demény et al. (2017). Moreover, new figures has been added as panels A and B to Fig. 1 showing the complementary characteristics of the stable isotope records of the two speleothems.

**However, my suggestion would be to analyse both records separately and then compare it to the composite. This alternative approach would highlight potential discrepancies between the records and would make the study more robust.**

**I wonder if the composite uncertainties have an effect on the periodogram used to decide the appropriate cut-off period if they are considered with an iterative approach. Do significant peaks remain significant? How do these uncertainties affect the spectra analysis of the records when it comes to interpreting in terms of climate variables? This could be also argued for a single speleothem record, as all proxy records have dating and measurement uncertainties.**

Thank you for the suggestion, due to the short time period covered by the VK1 speleothem (44 data points), the comparison of the rLSPs of the composite and the standalone records was only done using the NU2 stable isotope data (122 data points). The oxygen and carbon NU2 stable isotope records were processed the same way as the composite in the MS (spline interpolation, calculation of annual averages, spectral analysis). Moreover, the rLSPs of the original gappy NU2 stable isotope time series were estimated as well, to compare the rLSPs of both the interpolated and the raw data.

It was found that in the case of both oxygen (Fig. R3) and carbon (Fig. R4) stable isotope records in the period domain above the cut-off threshold

(i) the rLSPs of the raw and the interpolated NU2 stable isotope data show a similar pattern, i.e. the significant peaks are almost identical, especially in the period domain;

(ii) the significant periods in the composite's spectrum mirror the peaks in the original data's spectrum.

In addition, we can conclude that

(i) the composite record allowed to extend the spectral coverage to a period domain lower than the NU2 record by itself, because the two records complemented each other by filling their hiatuses (please see new Fig. 1a and the answer to your previous comment), thus ameliorating the sampling resolution.

(ii) the cut-off period determined for the composite record has proven to be accurate for the NU2 record as well, since it coherently captured the threshold below which discrepancies arose between the rLSPs of the NU2 record's interpolated and original raw data

(iii) merging replicated signals and the most definitely uncorrelated noise of multiple (in this case 2) proxies in a composite, is generally expected to improve the signal to noise ratio.

[Figure]

**Fig. R3: rLSPs of the original, and the spline interpolated $\Delta^{18}O_{Baradla}$ record of the Baradla speleothem (upper panel), and the original raw and the spline interpolated $\delta^{18}O$ record of the NU2 speleothem (lower panel). The bias-corrected 80% chi-squared limit of a fitted AR(1) process for the rLSPs is shown in grey, orange and green for the $\Delta^{18}O_{Baradla}$, the spline interpolated and the raw $\delta^{18}O$ record of the NU2 speleothem respectively. The vertical black line indicates the determined cut-off period.**

[Figure]

**Fig. R4: rLSPs of the original, and the spline interpolated $\Delta^{13}C_{Baradla}$ record of the Baradla speleothem (upper panel), and the original raw and the spline interpolated $\delta^{13}C$**

**record of the NU2 speleothem (lower panel). The bias-corrected 80% chi-squared limit of a fitted AR(1) process for the rLSPs is shown in grey and orange for the $\varDelta^{13}C_{Baradla}$, and the $\delta^{13}C$ record of the NU2 speleothem respectively. The vertical black line indicates the determined cut-off period.**

**I do not see any benefit in including the speleothems from Belgium in this study. They are just another example and they're only mentioned in the 25 lines of the discussion section, just before the conclusions. I suggest the authors either omit these records or discuss them more extensively.**

Accepted, thus Section 4 has been removed from the MS.

Minor comments

**L34: The temporal resolution of the speleothem proxy record will depend on the karst processes at play at that particular drip site. This can hugely vary from sub-annual to centennial resolution and therefore, it is not always possible to obtain a highly resolved temporal record. Regarding the "high spatial resolution"? I'd suggest that "they are well distributed worldwide" instead.**

We only said, that they can be sampled at high spatial and temporal resolution, but the sentence in question has been rephrased and extended as "…(iii) can be sampled at high spatial and temporal resolution depending on the growth rate of the speleothem, (iv) are well distributed worldwide, and (v) …" in lines 30-33

**Would it be possible to know what are the benefits of using a cubic spline in contrast to other interpolation methods?**

The Reviewer is referred to our answer given to her first general comment.

**L52: I don't understand the concept of "reference records". Why are these records considered to be a reference for their regions? This manuscript does not compare several series from the same region, so I think the usage of this term can become misleading. L52-53: An additional criteria to select "reference records" according to the authors, should be the overlap with meteo data (you could find accurately dated records close to annual that do not overlap with meteorological instrumental data).**

The paragraph in question has been rephrased and the problematic sentence removed.

**L86: There is no need to include the definition of delta here.**

Accepted and corrected, the definition has been removed and a short reference has been inserted into line 96 about how the isotope compositions are expressed.

**L92: "see section 2.1". That sentence is already part of section 2.1!**

The typo has been corrected.

**L106-109: It doesn't really matter for the purpose of this manuscript but you can obtain a longer PC-based NAO index from the 20CRv2c dataset, which goes back to 1851 (https://www.esrl.noaa.gov/psd/data/20thC_Rean/)**

Thank you for the comment.

**L117-118: I would like to see the periodogram of the original record along with the one from the reconstructed signal early in the text. In contrast, Figure 1 does not provide much information (it is difficult to visually compare both panels). This is related to my comment below for L134.**

The text has been rephrased in the corresponding line, because this was intended to be a general description. In addition the Figure, the Reviewer is specifically missing is Fig. 2 in the MS.

**L244-245: Which period corresponds to a "strong negative NAO mode" and for which frequency-band is this evident?**

The text has been extended in line 278 as "…(~1955 - ~1970; Fig. 5)…" and so signal can be detected in this time interval at any frequency.

**L134: I would like to see a figure showing the significant powers of both series (original and reconstructed)**

The Reviewer is referred to our answer given to her 6th General comment "However, my suggestion would be to analyse both records separately…", specifically to Figs. R3 & R4.

**The use of "composition" when referring to a composite series is confusing (for example "precipitation-composition" relationship in L212. Also, in L37, the usage of "composition" seems to mean "proxy records". Please clarify.**

Accepted and corrected in every necessary place

**L226: ". . .with the primary climate parameters", these being?**

Accepted and corrected, precipitation and temperature has been added to the sentence.

**Fig 5 (and others): could you please mark the cut-off periods below which the spectral analyses are not significant?**

The signal in those periods is not simply insignificant, but not present. These are now marked with dots on the relevant figures.